# Synthesized Policies for Transfer and Adaptation across Tasks and Environments

**Hexiang Hu** *
University of Southern California
Los Angeles, CA 90089
hexiangh@usc.edu

**Liyu Chen** *
University of Southern California
Los Angeles, CA 90089
liyuc@usc.edu

**Boqing Gong**
Tencent AI Lab
Bellevue, WA 98004
boqinggo@outlook.com

**Fei Sha** †
Netflix
Los Angeles, CA 90028
fsha@netflix.com

## Abstract

The ability to transfer in reinforcement learning is key towards building an agent of general artificial intelligence. In this paper, we consider the problem of learning to simultaneously transfer across both environments ($\varepsilon$) and tasks ($\tau$), probably more importantly, by learning from only sparse ($\varepsilon, \tau$) pairs out of all the possible combinations. We propose a novel compositional neural network architecture which depicts a meta rule for composing policies from environment and task embeddings. Notably, one of the main challenges is to learn the embeddings jointly with the meta rule. We further propose new training methods to disentangle the embeddings, making them both distinctive signatures of the environments and tasks and effective building blocks for composing the policies. Experiments on GRIDWORLD and THOR, of which the agent takes as input an egocentric view, show that our approach gives rise to high success rates on all the ($\varepsilon, \tau$) pairs after learning from only 40% of them.

## 1 Introduction

Remarkable progress has been made in reinforcement learning in the last few years [16, 21, 26]. Among these, an agent learns to discover its best policy of actions to accomplish a task, by interacting with the environment. However, the skills the agent learns are often tied for a specific pair of the environment ($\varepsilon$) and the task ($\tau$). Consequently, when the environment changes even slightly, the agent's performance deteriorates drastically [11, 28]. Thus, being able to swiftly adapt to new environments and transfer skills to new tasks is crucial for the agents to act in real-world settings.

*How can we achieve swift adaptation and transfer?* In this paper, we consider several progressively difficult settings. In the first setting, the agent needs to adapt and transfer to a new pair of environment and task, when the agent has been exposed to the environment and the task before (but not simultaneously). Our goal is to use as few as possible *seen* pairs (*i.e.*, a subset out of all possible ($\varepsilon$, $\tau$) combinations, as sparse as possible) to train the agent.

In the second setting, the agent needs to adapt and transfer across either environments or tasks, to those previously unseen by the agent. For instance, a home service robot needs to adapt from one home to another one but essentially accomplish the same sets of tasks, or the robot learns new tasks in the same home. In the third setting, the agent has encountered neither the environment nor the task

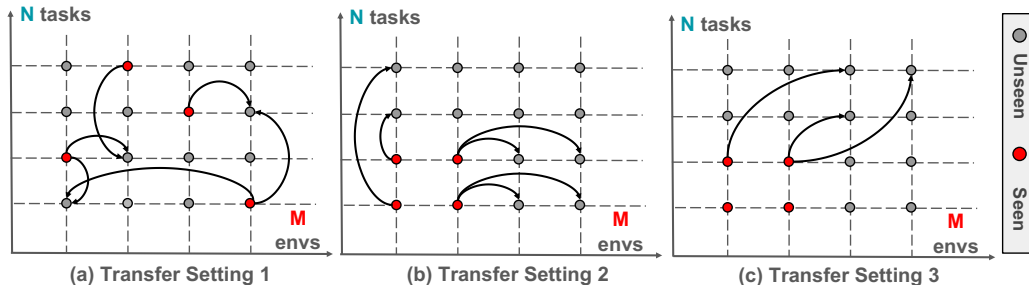

Figure 1: We consider a transfer learning scenario in reinforcement learning that considers transfer in both task and environment. Three different settings are presented here (see text for details). The **red dots** denote SEEN combinations, gray dots denote UNSEEN combinations, and arrows → denote transfer directions.

before. Intuitively, the second and the third settings are much more challenging than the first one and appear to be intractable. Thus, the agent is allowed to have a very limited amount of learning data in the target environment and/or task, for instance, from one demonstration, in order to transfer knowledge from its prior learning.

Figure 1 schematically illustrates the three settings. Several existing approaches have been proposed to address some of those settings [1–3, 14, 17, 24, 25]; for a detailed discussion, see related works in Section 2. A common strategy behind these works is to jointly learn through multi-task (reinforcement) learning [9, 18, 25]. Despite many progresses, however, adaptation and transfer remain a challenging problem in reinforcement learning where a powerful learning agent easily overfits to the environment or the task it has encountered, leading to poor generalization to new ones [11, 28].

In this paper, we propose a new approach to tackle this challenge. Our main idea is to learn a meta rule to synthesize policies whenever the agent encounters new environments or tasks. Concretely, the meta rule uses the embeddings of the environment and the task to compose a policy, which is parameterized as the linear combination of the policy basis. On the training data from seen pairs of environments and tasks, our algorithm learns the embeddings as well as the policy basis. For new environments or tasks, the agent learns the corresponding embeddings only while it holds the policy basis fixed. Since the embeddings are low-dimensional, a limited amount of training data in the new environment or task is often adequate to learn well so as to compose the desired policy.

While deep reinforcement learning algorithms are capable of memorizing and thus entangling representations of tasks and environments [28], we propose a disentanglement objective such that the embeddings for the tasks and the environments can be extracted to maximize the efficacy of the synthesized policy. Empirical studies demonstrate the importance of disentangling the representations.

We evaluated our approach on GRIDWORLD which we have created and the photo-realistic robotic environment THOR [13]. We compare to several leading methods for transfer learning in a significant number of settings. The proposed approach outperforms most of them noticeably in improving the effectiveness of transfer and adaptation.

## 2 Related Work

Multi-task [27] and transfer learning [24] for reinforcement learning (RL) have been long and extensively studied. Teh *et al*. [25] presented a distillation based method that transfers the knowledge from task specific agents to a multi-task learning agent. Andreas *et al*. [1] combined the option framework [23] and modular network [2], and presented an efficient multi-task learning approach which shares sub-policies across policy sketches of different tasks. Schaul *et al*. [19] encoded the goal state into value functions and showed its generalization to new goals. More recently, Oh *et al*. [17] proposed to learn a meta controller along with a set of parameterized policies to compose a policy that generalizes to unseen instructions. In contrast, we jointly consider the tasks and environments which can be both atomic, as we learn their embeddings without resorting to any external knowledge (e.g., text, attributes, etc.).

Several recent works [3, 6, 14, 29] factorize Q value functions with an environment-agnostic state-action feature encoding function and task-specific embeddings. Our model is related to this line of

work in spirit. However, as opposed to learning the value functions, we directly learn a factorized policy network with strengthened disentanglement between environments and tasks. This allows us to easily generalize better to new environments or tasks, as shown in the empirical studies.

## 3   Approach

We begin by introducing notations and stating the research problem formally. We then describe the main idea behind our approach, followed by the details of each component of the approach.

### 3.1   Problem Statement and Main Idea

**Problem statement.**   We follow the standard framework for reinforcement learning [22]. An agent interacts with an environment by sequentially choosing actions over time and aims to maximize its cumulative rewards. This learning process is abstractly described by a Markov decision process with the following components: a space of the agent's state $s \in \mathcal{S}$, a space of possible actions $a \in \mathcal{A}$, an initial distribution of states $p_0(s)$, a stationary distribution characterizing how the state at time $t$ transitions to the next state at $(t+1)$: $p(s_{t+1}|s_t, a_t)$, and a reward function $r := r(s, a)$.

The agent's actions follow a policy $\pi(a|s) : \mathcal{S} \times \mathcal{A} \to [0, 1]$, defined as a conditional distribution $p(a|s)$. The goal of the learning is to identify the optimal policy that maximizes the discounted cumulative reward: $R = \mathbb{E}[\sum_{t=0}^{\infty} \gamma^t r(s_t, a_t)]$, where $\gamma \in (0, 1]$ is a discount factor and the expectation is taken with respect to the randomness in state transitions and taking actions. We denote by $p(s|s', t, \pi)$ the probability at state $s$ after transitioning $t$ time steps, starting from state $s'$ and following the policy $\pi$. With it, we define the discounted state distribution as $\rho^\pi(s) = \sum_{s'} \sum_{t=1}^{\infty} \gamma^{t-1} p_0(s') p(s|s', t, \pi)$.

In this paper, we study how an agent learns to accomplish a variety of tasks in different environments. Let $\mathcal{E}$ and $\mathcal{T}$ denote the sets of the environments and the tasks, respectively. We assume the cases of finite sets but it is possible to extend our approach to infinite ones. While the most basic approach is to learn an optimal policy under each pair $(\varepsilon, \tau)$ of environment and task, we are interested in *generalizing to all combinations in $(\mathcal{E}, \mathcal{T})$, with interactive learning from a limited subset of $(\varepsilon, \tau)$ pairs*. Clearly, the smaller the subset is, the more desirable the agent's generalization capability is.

**Main idea.**   In the rest of the paper, we refers to the limited subset of pairs as *seen pairs* or *training pairs* and the rest ones as *unseen pairs* or *testing pairs*. We assume that the agent does *not* have access to the unseen pairs to obtain any interaction data to learn the optimal policies directly. In computer vision, such problems have been intensively studied in the frameworks of unsupervised domain adaptation and zero-shot learning, for example, [4, 5, 8, 15]. There are totally $|\mathcal{E}| \times |\mathcal{T}|$ pairs – our goal is to learn from $O(|\mathcal{E}| + |\mathcal{T}|)$ training pairs and generalize to all.

Our main idea is to synthesize policies for the unseen pairs of environments and tasks. In particular, our agent learns two sets of embeddings: one for the environments and the other for the tasks. Moreover, the agent also learns how to compose policies using such embeddings. Note that learning both the embeddings and how to compose happens on the training pairs. For the unseen pairs, the policies are constructed and used right away — if there is interaction data, the policies can be further fine-tuned. However, even without such interaction data, the synthesized policies still perform well.

To this end, we desire our approach to jointly supply two aspects: a compositional structure of **Synthesized Policies (SYNPO)** from environment and task embeddings and a disentanglement learning objective to learn the embeddings. We refer this entire framework as SYNPO and describe its details in what follows.

### 3.2   Policy Factorization and Composition

Given a pair $z = (\varepsilon, \tau)$ of an environment $\varepsilon$ and a task $\tau$, we denote by $e_\varepsilon$ and $e_\tau$ their embeddings, respectively. The policy is synthesized with a bilinear mapping

$$\pi_z(a|s) \propto \exp(\boldsymbol{\psi}_s^{\mathrm{T}} \boldsymbol{U}(e_\varepsilon, e_\tau) \boldsymbol{\phi}_a + b_\pi) \tag{1}$$

where $b_\pi$ is a scalar bias, and $\boldsymbol{\psi}_s$ and $\boldsymbol{\phi}_a$ are featurized states and actions (for instances, image pixels or the feature representations of an image). The bilinear mapping given by the matrix $\boldsymbol{U}$ is

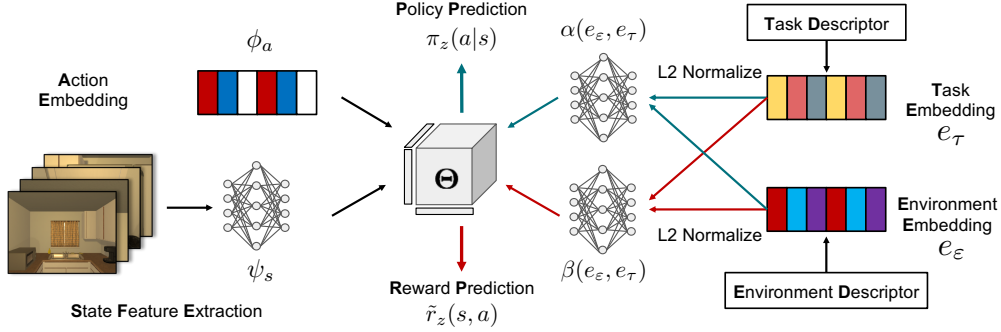

Figure 2: Overview of our proposed model. Given a task and an environment, the corresponding embeddings $e_\varepsilon$ and $e_\tau$ are retrieved to compose the policy coefficients and reward coefficients. Such coefficients then linearly combine the shared basis and synthesize a policy (and a reward prediction) for the agent.

parameterized as the linear combination of $K$ basis matrices $\boldsymbol{\Theta}_k$,

$$\boldsymbol{U}(e_\varepsilon, e_\tau) = \sum_{k=1}^{K} \alpha_k(e_\varepsilon, e_\tau)\boldsymbol{\Theta}_k. \tag{2}$$

Note that the combination coefficients depend on the specific pair of environment and task while the basis is *shared* across all pairs. They enable knowledge transfer from the seen pairs to unseen ones.

Analogously, during learning (to be explained in detail in the later section), we predict the rewards by modeling them with the same set of basis but different combination coefficients:

$$\tilde{r}_z(s, a) = \boldsymbol{\psi}_s^{\mathrm{T}}\boldsymbol{V}(e_\varepsilon, e_\tau)\boldsymbol{\phi}_a + b_r = \boldsymbol{\psi}_s^{\mathrm{T}}\left(\sum_k \beta_k(e_\varepsilon, e_\tau)\boldsymbol{\Theta}_k\right)\boldsymbol{\phi}_a + b_r \tag{3}$$

where $b_r$ is a scalar bias. Note that similar strategies for learning to predict rewards along with learning the policies have also been studied in recent works [3, 12, 29]. We find this strategy helpful too (cf. details in our empirical studies in Section 4).

Figure 2 illustrates the model architecture described above. In this paper, we consider agents that take egocentric views of the environment, so a convolutional neural network is used to extract the state features $\boldsymbol{\psi}_s$ (cf. the bottom left panel of Figure 2). The action features $\boldsymbol{\phi}_a$ are learned as a look-up table. Other model parameters include the basis $\boldsymbol{\Theta}$, the embeddings $e_\varepsilon$ and $e_\tau$ in the look-up tables respectively for the environments and the tasks, and the coefficient functions $\alpha_k(\cdot, \cdot)$ and $\beta_k(\cdot, \cdot)$ for respectively synthesizing the policy and reward predictor. The coefficient functions $\alpha_k(\cdot, \cdot)$ and $\beta_k(\cdot, \cdot)$ are parameterized with one-hidden-layer MLPs with the inputs being the concatenation of $e_\varepsilon$ and $e_\tau$, respectfully.

### 3.3 Disentanglement of the Embeddings for Environments and Tasks

In SYNPO, both the embeddings and the bilinear mapping are to be learnt. In an alternative but equivalent form, the policies are formulated as

$$\pi_z(a|s) \propto \exp\left(\sum_k \alpha_k(e_\varepsilon, e_\tau)\boldsymbol{\psi}_s^{\mathrm{T}}\boldsymbol{\Theta}_k\boldsymbol{\phi}_a + b_\pi\right). \tag{4}$$

As the defining coefficients $\alpha_k$ are parameterized by a neural network whose inputs and parameters are both optimized, we need to impose additional structures such that the learned embeddings facilitate the transfer across environments or tasks. Otherwise, the learning could overfit to the seen pairs and consider each pair in unity, thus leading to poor generalization to unseen pairs.

To this end, we introduce discriminative losses to distinguish different environments or tasks through the agent's trajectories. Let $\boldsymbol{x} = \{\boldsymbol{\psi}_s^{\mathrm{T}}\boldsymbol{\Theta}_k\boldsymbol{\phi}_a\} \in \mathbb{R}^K$ be the state-action representation. For the agent interacting with an environment-task pair $z = (\varepsilon, \tau)$, we denote its trajectory as $\{\boldsymbol{x}_1, \boldsymbol{x}_2, \cdots, \boldsymbol{x}_t, \ldots\}$. We argue that a good embedding (either $e_\varepsilon$ or $e_\tau$) ought to be able to tell from which environment or

task the trajectory is from. In particular, we formulate this as a multi-way classification where we desire $\boldsymbol{x}_t$ (on average) is telltale of its environment $\varepsilon$ or task $\tau$:

$$\ell_\varepsilon := -\sum_t \log P(\varepsilon|\boldsymbol{x}_t) \text{ with } P(\varepsilon|\boldsymbol{x}_t) \propto \exp\left(g(\boldsymbol{x}_t)^\mathrm{T} e_\varepsilon\right) \tag{5}$$

$$\ell_\tau := -\sum_t \log P(\tau|\boldsymbol{x}_t) \text{ with } P(\tau|\boldsymbol{x}_t) \propto \exp\left(h(\boldsymbol{x}_t)^\mathrm{T} e_\tau\right) \tag{6}$$

where we use two nonlinear mapping functions ($g(\cdot)$ and $h(\cdot)$, parameterized by one-hidden-layer MLPs) to transform the state-action representation $\boldsymbol{x}_t$, such that it retrieves $e_\varepsilon$ and $e_\tau$. These two functions are also learnt using the interaction data from the seen pairs.

## 3.4 Learning

Our approach (SYNPO) relies on the modeling assumption that the policies (and the reward predicting functions) are factorized in the axes of the environment and the task. This is a generic assumption and can be integrated with many reinforcement learning algorithms. In this paper, we study its effectiveness on imitation learning (mostly) and also reinforcement learning.

In imitation learning, we denote by $\pi_z^e$ the expert policy of combination $z$ and apply the simple strategy of "behavior cloning" with random perturbations to learn our model from the expert demonstration [10]. We employ a cross-entropy loss for the policy as follows:

$$\ell_{\pi_z} := -\mathbb{E}_{s\sim\rho^{\pi_z^e}, a\sim\pi_z^e}[\log \pi_z(a|s)]$$

A $\ell_2$ loss is used for learning the reward prediction function, $\ell_{r_z} := \mathbb{E}_{s\sim\rho^{\pi_z^e}, a\sim\pi_z^e}\|\tilde{r}_z(s,a) - r_z(s,a)\|_2$. Together with the disentanglement losses, they form the overall loss function

$$\mathcal{L} := \mathbb{E}_z[\ell_{\pi_z} + \lambda_1 \ell_{r_z} + \lambda_2 \ell_\varepsilon + \lambda_3 \ell_\tau]$$

which is then optimized through experience replay, as shown in **Algorithm 1** in the supplementary materials (Suppl. Materials). We choose the value of those hyper-parameters $\lambda_i$ so that the contributions of the objectives are balanced. More details are presented in the Suppl. Materials.

## 3.5 Transfer to Unseen Environments and Tasks

Eq. 1 is used to synthesize a policy for any $(\varepsilon, \tau)$ pair, as long as the environment and the task — not necessarily the pair of them — have appeared at least once in the training pairs. If, however, a new environment and/or a new task appears (corresponding to the transfer setting 2 or 3 in Section 1), fine-tuning is required to extract their embeddings. To do so, we keep all the components of our model fixed except the look-up tables (i.e., embeddings) for the environment and/or the task. This effectively re-uses the policy composition rule and enables fast learning of the environment and/or the task embeddings, after seeing a few number of demonstrations. In the experiments, we find it works well even with only one shot of the demonstration.

# 4 Experiments

We validate our approach (SYNPO) with extensive experimental studies, comparing with several baselines and state-of-the-art transfer learning methods.

## 4.1 Setup

We experiment with two simulated environments[3]: GRIDWORLD and THOR [13], in both of which the agent takes as input an egocentric view (cf. Figure 3). Please refer to the Suppl. Materials for more details about the state feature function $\psi_s$ used in these simulators.

**GRIDWORLD and tasks.** We design twenty $16 \times 16$ grid-aligned mazes, some of which are visualized in Figure 3 (a). The mazes are similar in appearance but differ from each other in topology. There are five colored blocks as "treasures" and the agent's goal is to collect the treasures in pre-specified orders, *e.g.*, *"Pick up Red and then pick up Blue"*. At a time step, the "egocentric" view

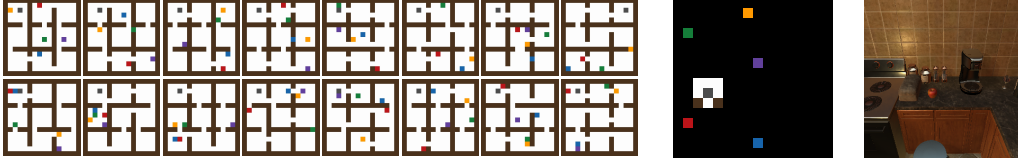

Figure 3: From left to right: (a) Some sample mazes of our GRIDWORLD dataset. They are similar in appearance but different in topology. Demonstrations of an agent's egocentric views of (b) GRIDWORLD and (c) THOR.

observed by the agent consists of the agent's surrounding within a $3 \times 3$ window and the treasures' locations. At each run, the locations of the agent and treasures are randomized. We consider twenty tasks in each environment, resulting $|\mathcal{E}| \times |\mathcal{T}| = 400$ pairs of $(\varepsilon, \tau)$ in total. In the transfer setting 1 (cf. Figure 1(a)), we randomly choose 144 pairs as the training set under the constraint that each of the environments appears at least once, so does any task. The remaining 256 pairs are used for testing. For the transfer settings 2 and 3 (cf. Figure 1(b) and (c)), we postpone the detailed setups to Section 4.2.2.

**THOR [13] and tasks.** We also test our method on THOR, a challenging 3D simulator where the agent is placed in indoor photo-realistic scenes. The tasks are to search and act on objects, *e.g.*, *"Put the cabbage to the fridge"*. Different from GRIDWORLD, the objects' locations are unknown so the agent has to search for the objects of interest by its understanding of the visual scene (cf. Figure 3(c)). There are 7 actions in total (*look up*, *look down*, *turn left*, *turn right*, *move forward*, *open/close*, *pick up/put down*). We run experiments with 19 scenes $\times$ 21 tasks in this simulator.

**Evaluations.** We evaluate the agent's performance by the averaged success rate (AvgSR.) for accomplishing the tasks, limiting the maximum trajectory length to 300 steps. For the results reported in numbers (*e.g.*, Tables 1), we run 100 rounds of experiments for each $(\varepsilon, \tau)$ pair by randomizing the agent's starting point and the treasures' locations. To plot the convergence curves (*e.g.*, Figure 4), we sample 100 $(\varepsilon, \tau)$ combinations and run one round of experiment for each to save computation time. We train our algorithms under 3 random seeds and report the mean and standard deviation (std).

**Competing methods.** We compare our approach (SYNPO) with the following baselines and competing methods. Note that our problem setup is new, so we have to adapt the competing methods, which were proposed for other scenarios, to fit ours.

- **MLP.** The policy network is a multilayer perceptron whose input concatenates state features and the environment and task embeddings. We train this baseline using the proposed losses for our approach, including the disentanglement losses $\ell_\epsilon, \ell_\tau$; it performs worse without $\ell_\epsilon, \ell_\tau$.
- **Successor Feature (SF).** We learn the *successor feature* model [3] by Q-imitation learning for fair comparison. We strictly follow [14] to set up the learning objectives. The key difference of SF from our approach is its lack of capability in capturing the environmental priors.
- **Module Network (ModuleNet).** We also implement a module network following [7]. Here we train an environment specific module for each environment and a task specific module for each task. The policy for a certain $(\varepsilon, \tau)$ pair is assembled by combining the corresponding environment module and task module.
- **Multi-Task Reinforcement Learning (MTL).** This is a degenerated version of our method, where we ignore the distinctions of environments. We simply replace the environment embeddings by zeros for the coefficient functions. The disentanglement loss on task embeddings is still used since it leads to better performances than otherwise.

Please refer to the Suppl. Materials for more experimental details, including all the twenty GRID-WORLD mazes, how we configure the rewards, optimization techniques, feature extraction for the states, and our implementation of the baseline methods.

## 4.2 Experimental Results on GRIDWORLD

We first report results on the adaptation and transfer learning setting 1, as described in Section 1 and Figure 1(a). There, the agent acts upon a new pair of environment and task, both of which it has

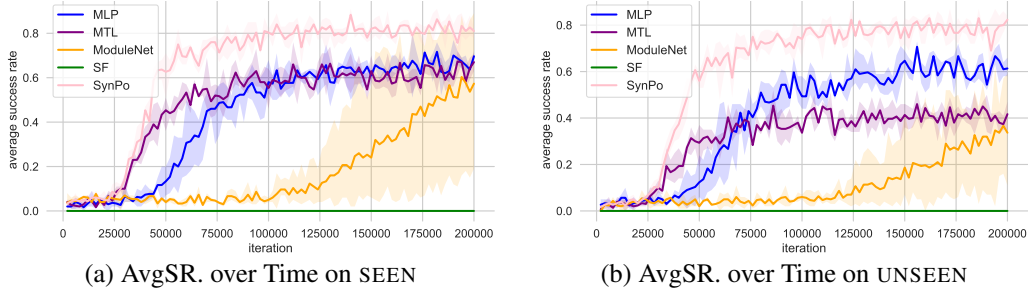

(a) AvgSR. over Time on SEEN  (b) AvgSR. over Time on UNSEEN

Figure 4: **On GRIDWORLD.** Averaged success rate (AvgSR) on SEEN pairs and UNSEEN pairs, respectively. Results are reported with $|\mathcal{E}| = 20$ *and* $|\mathcal{T}| = 20$. We report mean and std based on 3 training random seeds.

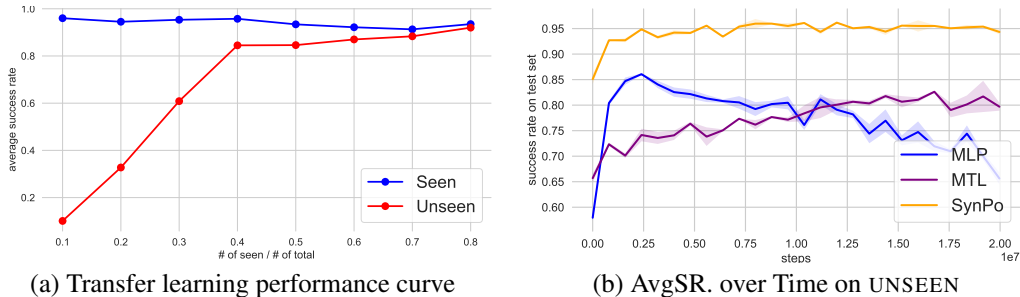

(a) Transfer learning performance curve  (b) AvgSR. over Time on UNSEEN

Figure 5: (a) **Transfer learning performance** (in AvgSR.) with respect to the ratio: # SEEN pairs / # TOTAL pairs, with $|\mathcal{E}| = 10$ *and* $|\mathcal{T}| = 10$. (b) **Reinforcement learning performance** on unseen pairs of different approaches (with PPO [20]). MLP overfits, MTL improves slightly, and SYNPO achieves 96.16% AvgSR.

encountered during training but not in the same $(\varepsilon, \tau)$ pair. The goal is to use as sparse $(\varepsilon, \tau)$ pairs among all the combinations as possible to learn and yet still able to transfer successfully.

### 4.2.1 Transfer to Previously Encountered Environments and Tasks

**Main results.** Table 1 and Figure 4 show the success rates and convergence curves, respectively, of our approach and the competing methods averaged over the seen and unseen $(\varepsilon, \tau)$ pairs. SYNPO consistently outperforms the others in terms of both the convergence and final performance, by a significant margin. On the seen split, MTL and MLP have similar performances, while MTL performs worse comparing to MLP on the unseen split (*i.e.* in terms of the generalization performance), possibly because it treats all the environments the same.

We design an extreme scenario to further challenge the environment-agnostic methods (e.g., MTL). We reduce the window size of the agent's view to one, so the agent sees the cell it resides and the treasures' locations and nothing else. As a result, MTL suffers severely, MLP performs moderately well, and SYNPO outperforms both significantly *(unseen AvgSR: MTL=6.1%, MLP=66.1%, SYNPO = 76.8%)*. We conjecture that the environment information embodied in the states is crucial for the agent to beware of and generalize across distinct environments. More discussions are deferred to the Suppl. Materials.

**How many seen $(\varepsilon, \tau)$ pairs do we need to transfer well?** Figure 5(a) shows that, not surprisingly, the transfer learning performance increases as the number of seen pairs increases. The acceleration slows down after the seen/total ratio reaches 0.4. In other words, when there is a limited budget, our approach enables the agent to learn from 40% of all possible $(\varepsilon, \tau)$ pairs and yet generalize well across the tasks and environments.

**Does reinforcement learning help transfer?** Beyond imitation learning, we further study our SYNPO for reinforcement learning (RL) under the same transfer learning setting. Specifically, we use PPO [20] to fine-tune the three top performing algorithms on GRIDWORLD. The results averaged over 3 random seeds are shown in Figure 5(b). We find that RL fine-tuning improves the transfer

Table 1: Performance (AvgSR.) of each method on GRIDWORLD (SEEN/UNSEEN = 144/256).

| Method | SF | ModuleNet | MLP | MTL | SYNPO |
|---|---|---|---|---|---|
| AvgSR. (SEEN) | $0.0 \pm 0.0\%$ | $50.9 \pm 33.8\%$ | $69.0 \pm 2.0\%$ | $64.1 \pm 1.2\%$ | $\mathbf{83.3 \pm 0.5\ \%}$ |
| AvgSR. (UNSEEN) | $0.0 \pm 0.0\%$ | $30.4 \pm 20.1\%$ | $66.1 \pm 2.6\%$ | $41.5 \pm 1.4\%$ | $\mathbf{82.1 \pm 1.5\%}$ |

Table 2: Performance of transfer learning in the settings 2 and 3 on GRIDWORLD

| Setting | Method | Cross Pair ($Q$'s $\varepsilon$, $P$'s $\tau$) | Cross Pair ($P$'s $\varepsilon$, $Q$'s $\tau$) | $Q$ Pairs |
|---|---|---|---|---|
| Setting 2 | MLP | 13.8% | 20.7% | 6.3% |
|  | SYNPO | **50.5%** | **21.5%** | **13.5%** |
| Setting 3 | MLP | 14.6% | 18.3% | 7.2% |
|  | SYNPO | **42.7%** | **19.4%** | **12.9%** |

performance for all the three algorithms. In general, MLP suffers from over-fitting, MTL is improved moderately yet with a significant gap to the best result, and SYNPO achieves the best AvgSR, **96.16%**.

**Ablation studies.** We refer readers to the Suppl. Materials for ablation studies of the learning objectives.

#### 4.2.2 Transfer to Previously Unseen Environments or Tasks

Now we investigate how effectively one can schedule transfer from seen environments and tasks to unseen ones, i.e., the settings 2 and 3 described in Section 1 and Figure 1(b) and (c). The seen pairs (denoted by $P$) are constructed from ten environments and ten tasks; the remaining ten environments and ten tasks are unseen (denoted by $Q$). Then we have two settings of transfer learning.

One is to transfer to pairs which cross the seen set $P$ and unseen set $Q$ – this corresponds to the setting 2 as the embeddings for either the unseen tasks or the unseen environments need to be learnt, but not both. Once these embeddings are learnt, we use them to synthesize policies for the test $(\varepsilon, \tau)$ pairs. This mimics the style *"incremental learning of small pieces and integrating knowledge later"*.

The other is the transfer setting 3. The agent learns policies via learning embeddings for the tasks and environments of the unseen set $Q$ and then composing, as described in section 3.5. Using the embeddings from $P$ and $Q$, we can synthesize policies for any $(\varepsilon, \tau)$ pair. This mimics the style of *"learning in giant jumps and connecting dots"*.

**Main results.** Table 2 contrasts the results of the two transfer learning settings. Clearly, setting 2 attains stronger performance as it "incrementally learns" the embeddings of either the tasks or the environments but not both, while setting 3 requires learning both simultaneously. It is interesting to see this result aligns with how effective human learns.

Figure 6 visualizes the results whose rows are indexed by tasks and columns by environments. The seen pairs in $P$ are in the upper-left quadrant and the unseen set $Q$ is on the bottom-right. We refer readers to the Suppl. Materials for more details and discussions of the results.

### 4.3 Experimental Results on THOR

**Main results.** The results on the THOR simulator are shown in Table 3, where we report our approach as well as the top performing ones on GRIDWORLD. Our SYNPO significantly outperforms three competing ones for both seen pairs and unseen pairs. Moreover, our approach also has the best performance of success rate on seen to unseen, indicating that it is less prone to overfitting than the other methods. More details are included in the Suppl. Materials.

## 5 Conclusion

In this paper, we consider the problem of learning to simultaneously transfer across both environments ($\varepsilon$) and tasks ($\tau$) under the reinforcement learning framework and, more importantly, by learning from only sparse $(\varepsilon, \tau)$ pairs out of all the possible combinations. Specifically, we present a novel approach that learns to synthesize policies from the disentangled embeddings of environments and tasks. We evaluate our approach for the challenging transfer scenarios in two simulators, GRID-

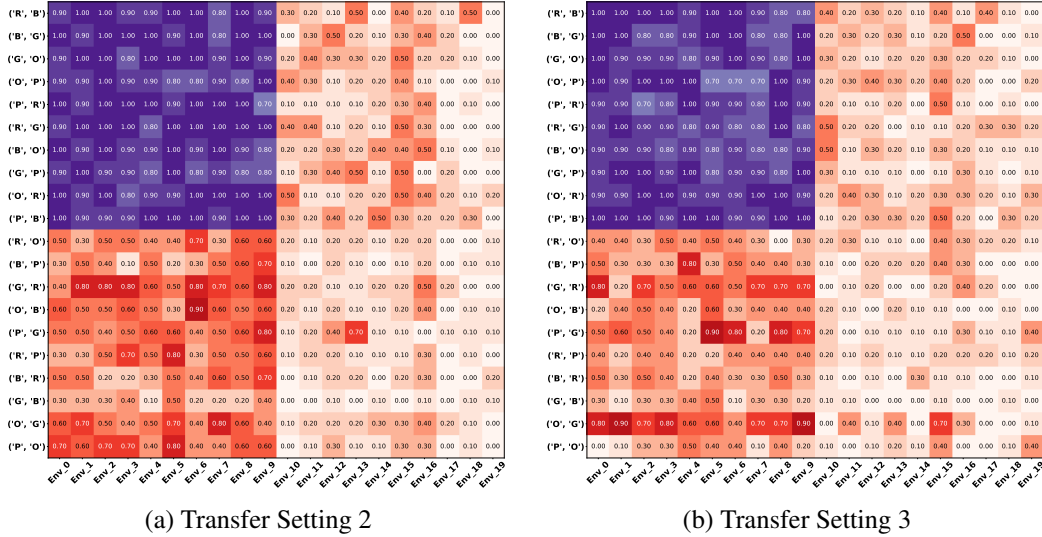

|                          | (a) Transfer Setting 2 | (b) Transfer Setting 3 |

Figure 6: Transfer results of settings 2 and 3. AvgSRs are marked in the grid (see Suppl. Materials for more visually discernible plots). The tasks and environments in the **purple cells** are from the unseen $Q$ set and the **red cells** correspond to the rest. Darker color means better performance. It shows that cross-task transfer is easier than cross-environment.

Table 3: Performance of each method on THOR (SEEN/UNSEEN=144/199)

| Method | ModuleNet | MLP | MTL | SYNPO |
|---|---|---|---|---|
| AvgSR. (SEEN) | 51.5 % | 47.5% | 52.2% | **55.6%** |
| AvgSR. (UNSEEN) | 14.4 % | 25.8% | 33.3% | **35.4%** |

WORLD and THOR. Empirical results verify that our method generalizes better across environments and tasks than several competing baselines.

**Acknowledgments** We appreciate the feedback from the reviewers. This work is partially supported by DARPA# FA8750-18-2-0117, NSF IIS-1065243, 1451412, 1513966/ 1632803/1833137, 1208500, CCF-1139148, a Google Research Award, an Alfred P. Sloan Research Fellowship, gifts from Facebook and Netflix, and ARO# W911NF-12-1-0241 and W911NF-15-1-0484.

## Footnotes

† On leave from University of Southern California (feisha@usc.edu).

[3]The implementation of the two simulated environments are available on `https://www.github.com/sha-lab/gridworld` and `https://www.github.com/sha-lab/thor`, respectfully.

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
