[Supplementary Material · supplementary-material.pdf]

# Synthesized Policies for Transfer and Adaptation across Tasks and Environments

## Supplementary Material

**Hexiang Hu** *
University of Southern California
Los Angeles, CA 90089
hexiangh@usc.edu

**Liyu Chen** *
University of Southern California
Los Angeles, CA 90089
liyuc@usc.edu

**Boqing Gong**
Tencent AI Lab
Bellevue, WA 98004
boqinggo@outlook.com

**Fei Sha** †
Netflix
Los Angeles, CA 90028
fsha@netflix.com

In this Supplementary Material, we provide details omitted in the main paper:

- Section 1: Detailed configurations about GRIDWORLD and THOR simulators.
- Section 2: Imitation learning algorithm and optimization details.
- Section 3: Reinforcement learning algorithm and optimization details.
- Section 4: Implementation details about SynPo and baselines.
- Section 5: Additional experimental results to the main text.

## 1 Details on simulators

### 1.1 Details about GRIDWORLD Configurations

As we have mentioned in the main text, there are in total 20 environments for this simulator, which we listed as Figure 9. The tasks presented in this simulator includes a sequential execution of picking up two treasures in different colors. The agent can observe the layout of the environment inside a 3x3 square centered at the agent's current position (see Figure 1 (a) for details). The agent can take 5 actions, which includes moving in the four directions and picking up an object right below it. Note that in each run of a certain given task, the locations of both agent and treasures are randomized.

In terms of the reward setting, we follow the common practice and set the reward for moving one step to be -0.01 and touching a wall to be an additional - 0.01. Picking up a target treasure gives 1 unit of the reward and completing a task gives 10 unites of the reward. Picking up a wrong target directly ends an episode and gives reward -10. During the training, we use an optimal planner with shortest path search algorithm for expert policy. To represent a state for our network, we follow the practice in DQN [6] and concatenate the last four observations as the input to the policy.

### 1.2 Details about THOR Configurations.

THOR [5] is a 3D robotic simulator developed recently for simulating the indoor environments a robot could encounter. An agent is working like a real robot with a first-person view camera, which delivers RGB images in egocentric view (see Figure 1 (b) for details). The environment has interactable components that a agent can play with, which enables the learning of human like behaviors such as

|  (a) Agent's View in GRIDWORLD | (b) Egocentric View in THOR |

Figure 1: Demonstrations of agent's view in two simulators. In the left, we present the agent's input state of GRIDWORLD. An agent only have the vision to its surrounding context and the locations of all treasures (see (a)). Similarly, in the THOR, an agent has access to an egocentric image that represents the first-person viewpoint (see (b)).

semantic planning [9] and indoor navigation [10]. We describe the concrete settings we used as what follows.

We extract the image features using convolutional neural networks to represent an observation for each egocentric view of a robotic agent. Specifically, we extract the activation output from the penultimate layer of a Resnet101 [3] pre-trained on ImageNet [1], which has the dimensionality of 2048. Similar to the GRIDWORLD experiments, we then concatenate those features of the last four observations as the input to the policy network. The agent can take 7 actions in THOR: move ahead, turn left, turn right, look up, look down, open/close an object, pick up/put down an object. We set the reward for moving one step to be -0.01 and executing invalid actions to be -0.01. The reward of picking up the correct object is 1, and the reward of finishing the task is 10. Picking up the wrong object and putting the object in the wrong receptacle ends an episode and gives -10 units of the reward. The interactable objects, receptacles and index of environments (kitchens) are listed in table 1. In our experiment we selected environments with similar size (see Table 1 for the complete list).

Table 1: interactable objects, receptacles and environment indexes in THOR

| Entries | Values |
| --- | --- |
| Objects | Container, Lettuce, Mug, Tomato, Plate, Apple, Bowl |
| Receptacles | Fridge, Microwave, Sink |
| Environments | Kitchen {1, 2, 3, 4, 5, 6, 8, 9, 11, 12, 18, 22, 23, 24, 25, 27, 28, 20, 30} |

## 2 Imitation Learning Algorithm and Optimization Details

As mentioned in the main text, now we describe the imitation learning algorithm used for learning SYNPO and all baseline models. The concrete details are presented in Algorithm 1.

In each episode, we sample a trajectory using the expert policy and store it into the replay buffer. At the end of each episode, we sample 64 trajectories uniformly from the replay buffer to calculate the total loss. Here, the size of replay buffer for storing expert trajectories is 20,000. In each episode, we uniformly sample 64 trajectories from the replay buffer (coming from different $\varepsilon$ and $\tau$ pairs) to compute the loss. We set the hyper-parameters $\lambda$ as follows: there is $\lambda_1 = 0.01$ for reward prediction; $\lambda_2 = 0.1$ and $\lambda_3 = 0.001$ for environment and task disentanglement loss. The dimensionality of environment embedding and task embedding are 128. Besides, we use Adam [4] as the optimizer with the initial learning rate set to be 0.001. Additionally, we set the value of weight decay factor to be 0.001 in all our experiments.

**Algorithm 1** Policy Imitation Learning Algorithm.

---

**Input:** Given training simulators $\mathbf{simulator}(z)$, where $z \in (\mathcal{E}, \mathcal{T})_{\mathtt{train}}$
Initialize Expert Replay Memory $\mathcal{D}^{\mathbf{E}}$ with the capacity $\mathbf{N}$
**for** episode = 1, M **do**
   Sample $z \in (\mathcal{E}, \mathcal{T})_{\mathtt{train}}$
   $\mathrm{TRAJ}_z(\{s_i, a_i, r_i\}; \pi^{\mathbf{E}}) = \mathrm{ROLLOUT}\Big(\pi_z^{\mathbf{E}}, \mathbf{simulator}(z)\Big)$
   Store $\mathrm{TRAJ}_z(\{s_i, a_i, r_i\}; \pi^{\mathbf{E}})$ to $\mathcal{D}^{\mathbf{E}}$
   Sample a random mini-batch $\mathbf{B}$ with $|\mathbf{B}|$ trajectories from $\mathcal{D}^{\mathbf{E}}$
   Compute gradient $\nabla \mathcal{L}$ and update the parameters with specified optimizer
**end for**

---

# 3 Reinforcement Learning Algorithm and Optimization Details

As mentioned in the main text, we have employed reinforcement learning to further fine-tune our model, which archived improvement in transfer learning performances. Now we describe the detailed setups of our experiments. We use PPO [8] to fine-tune our model. We optimize our model by RMSProp with learning rate $0.000025$ and weight decay $0.0001$. We use GAE [7] to calculate advantages, with $\gamma = 0.99$ and $\lambda = 0.95$, entropy weight is $0.01$, rollout length $128$, objective clipping ratio $0.1$. Gradient norms are clipped to $0.5$. We divide the trajectories collected into 4 mini batches and do four optimization steps on each update. We fine-tuned our model for $2 \times 10^7$ steps. During RL fine-tuning we also included our disentangling objectives as auxilary loss.

# 4 Implementation Details

## 4.1 Details about our Policy Network for SYNPO in GRIDWORLD

First, we introduce the specific setups we used for policy networks in GRIDWORLD. We directly parameterize the outcome of a dot product between $\mathbf{\Theta}$ and $\phi_a$ as a tensor, for the sake of computation efficiency in practice. However, our model, as mentioned in the main text, is indeed a bilinear policy. Therefore, with a more general application scenario that action space ($|\mathcal{A}|$) is large, we can apply the original form of our approach and learn separate action embeddings $\phi_a$ with the shared basis $\mathbf{\Theta}$. The coefficient functions $\alpha(\cdot)$ and $\beta(\cdot)$ that compose environment and task embeddings are one-hidden-layer MLPs with 512 hidden units and output size of 128. The dimension of the state feature $\psi_s$ extracted from ResNet before the bilinear weight $U$ is 128. The state feature extractor is a customized ResNet. Its concrete structure is shown as Table 2. The dimensionality of the environment embeddings $e_\varepsilon$ and task embeddings $e_\tau$ are 128.

Table 2: Structure of State Feature Function $\psi_s$ in GRIDWORLD

| group name | output size | block type | stride |
|:---:|:---:|:---:|:---:|
| input | $16 \times 16 \times 3$ | - | - |
| conv 1 | $8 \times 8 \times 32$ | $\begin{bmatrix} 3 \times 3, 32 \\ 3 \times 3, 32 \end{bmatrix} \times 2$ | 2 |
| conv 2 | $4 \times 4 \times 64$ | $\begin{bmatrix} 3 \times 3, 64 \\ 3 \times 3, 64 \end{bmatrix} \times 2$ | 2 |
| conv 3 | $2 \times 2 \times 128$ | $\begin{bmatrix} 3 \times 3, 128 \\ 3 \times 3, 128 \end{bmatrix} \times 2$ | 2 |
| conv 4 | $2 \times 2 \times 256$ | $\begin{bmatrix} 3 \times 3, 256 \\ 3 \times 3, 256 \end{bmatrix} \times 2$ | 0 |
| fc | 128 | $\begin{bmatrix} 1024 \times 128 \end{bmatrix}$ | - |

## 4.2 Details about our Policy Network for SYNPO in THOR

Next we describe the network setups we used in THOR. Again, we directly parameterize the outcome of a dot product between $\Theta$ and $\phi_a$ as a tensor, as the action space is small ($|\mathcal{A}| = 7$) in this simulator. With the stacked $2,048 \times 4$ dimensional ResNet101 feature as input, we learn a two 1-D convolutional networks with kernel size of 3 and stride of 2, which first reduces the dimensionality of feature to 1,024 and then aggregates over the temporal axis. Next, the encoding of visual feature is then concatenated with an embedding ($e_{\mathbf{obj}}$) that represents object the agent is carrying. The concatenated feature vector is next input into a one-hidden-layer MLP wth hidden state of 2,048 dimension. The output of this MLP (which is also the final output ofstate feature function $\psi_s$) has dimension of 256. The concrete config is shown as Table 3. The dimensionality of the environment embeddings $e_\varepsilon$ and task embeddings $e_\tau$ are 128.

Table 3: Structure of State Feature Function $\psi_s$ in THOR

| group name | output size | block type | stride |
|---|---|---|---|
| image input | $2048 \times 4$ | - | - |
| conv 1 | $1024 \times 2$ | $\begin{bmatrix} 3 \times 1, 1024 \end{bmatrix}$ | 2 |
| conv 2 | $1024 \times 1$ | $\begin{bmatrix} 3 \times 1, 1024 \end{bmatrix}$ | 2 |
| concat | 1056 | concat $e_{\mathbf{obj}}$ | - |
| fc1 | 2048 | $\begin{bmatrix} 1056 \times 2048 \end{bmatrix}$ | - |
| fc2 | 256 | $\begin{bmatrix} 2048 \times 256 \end{bmatrix}$ | - |

## 4.3 Details about learning Disentanglement Objective

In addition to both of the above settings, we applied another set of one-hidden-layer MLPs $f_\varepsilon$ and $f_\tau$ (hidden=512) to represent the auxiliary function that project the high-dimensional trajectory feature $x$ to the embedding spaces $e_\varepsilon$ and $e_\tau$. Note that this function is only used in the disentanglement objective, and could be discarded during the deployment of policy network.

# 5 Additional Experimental Results

## 5.1 Complete Details of Main Results and Comparison between Methods

As mentioned in the main text, we put our complete results of GRIDWORLD here. Now we report not only the average success rate (AvgSR.) but also average reward (AvgReward), on both seen and unseen pairs.

Table 4: Performance of the best model for each method on GRIDWORLD (Seen/Unseen=144/256). All algorithms are trained using three random seeds and reported with mean and std. on each $(\varepsilon, \tau)$ pair, we sample the locations of agent and treasures for 100 times to evaluate the performances.

| Method | SF | ModuleNet | MLP | MTL | SYNPO |
|---|---|---|---|---|---|
| AvgSR. (SEEN) | $0.0 \pm 0.0\%$ | $50.9 \pm 33.8\%$ | $69.0 \pm 2.0\%$ | $64.1 \pm 1.2\%$ | $\mathbf{83.3 \pm 0.5\ \%}$ |
| AvgSR. (UNSEEN) | $0.0 \pm 0.0\%$ | $30.4 \pm 20.1\%$ | $66.1 \pm 2.6\%$ | $41.5 \pm 1.4\%$ | $\mathbf{82.1 \pm 1.5\%}$ |

We found that the trend of average reward on seen and unseen splits are quite similar to the trend of average success rate. We also note that the reward for successor feature (SF) is stable around -3, which indicated that the agent only tries to avoid negative reward and refuse to learn getting positive reward. On the contrary, all methods that make progress later starts with a lower average reward, meaning that the agent tries to complete the task by picking up objects but failed a lot at the beginning.

Figure 2: **Results on GRIDWORLD.** (a)-(b): Comparison between average success rate (ASR.) of algorithms on seen split and unseen split. (c)-(d): Comparison between average accumulated reward (AvgReward.) of algorithms in each episode on seen split and unseen split. Results are reported on the setting with $|\mathcal{E}| = 20$ *and* $|\mathcal{T}| = 20$. For each intermediate performance, we sample 100 $(\varepsilon, \tau)$ combinations and test one configuration to evaluate the performances. We evaluate models trained with 3 random seeds and report results in terms of the mean AvgSR and its standard deviation.

Figure 3: **An ablation study about our learning objectives.** We report the results of the ablated versions without the disentanglement loss (Disentg) on environment (EnvDisentg) and on task (TaskDisentg). (a)-(b): Comparison between average success rate (ASR.) of algorithms on SEEN split and UNSEEN split. (c)-(d): Comparison between average accumulated reward (AvgReward.) of algorithms in each episode on SEEN split and UNSEEN split. Results are reported on the setting with $|\mathcal{E}| = 20$ *and* $|\mathcal{T}| = 20$. Similarly, for each intermediate performance, we sample 100 $(\varepsilon, \tau)$ combinations to evaluate the performances.

Specifically, we find that SYNPO is consistently performing better across all metrics, in terms of both the convergence and final performance. On the seen splits, MTL and MLP have similar performances, while MTL has a much worse generalization performance on unseen splits, comparing to MLP, possibly due to over-fitting or the lack of the capability in recognizing environments. At the same time, it is worth noting that Module Network has a significantly larger variance in its performances, comparing against all other approaches. This is possibly due to the fact that the environment modules and task modules are adhered together during the inference, where instability could occur. Similar issue has also been reported by Devin *et al*. [2]. In addition, even in the best performing cases, ModuleNet could achieve a similar performances comparing to MLP and still far from approaching SYNPO's performance.

## 5.2    Ablation Studies of the Learning Objectives

How does each component in the objective function of our approach affect the performance of our model? Figure 3 shows that the task disentanglement loss is crucial for achieving good success rates on either seen or unseen pairs. This is probably because the differences between tasks are very subtle, making the agent hard to find the right distinct embeddings for them without the explicit task disentanglement loss. In contrast, the approach without the environment disentanglement loss can still reach a high success rate though it converges a bit slower.

## 5.3    Details on Transfer Learning Experiments

As mentioned in the main text, here we include the complete splits for the transfer learning study (Experiments evaluated the transfer learning result w.r.t. ratio # of seen vs.# of total). The success rate of our method on each pair is marked on the matrices. The full success rate matrices are shown as Figure 4 and Figure 5.

Specifically we case study the situation when this ratio is $0.2$. The detailed transfer learning performance is shown as Figure 6. Here each row corresponds to a task and each column corresponds to an environment. The red grids represents the unseen pairs and the purple grids represents the seen pairs. We mark the average success rate (over 100 runs of evaluations) in the grid to better quantitatively identify the performance at a pair of $(\varepsilon, \tau)$. The darker the color of a grid is, the better the corresponding performance. We can see that with the row "(O, R)" and column "env_0", although only entry along the row and column is seen by the model, the transfer learning performance does not fail completely. Instead, many entries along the row and column have a superior success rate. This supports our claim about disentanglement of the environment and task embedding, and at the same time indicates the success in the learning compositionality.

## 5.4    Details on Experiments of transfer setting 2 and setting 3

In this section, we describe the details of transfer learning settings. In both the setting 2 of "Incremental learning of small pieces and integrating knowledge later" and setting 3 of "Learning in giant jumps and connecting dots", we fix all parameters of the policy basis pre-trained on $P$ and fine-tune the network to learn new (randomly initialized) embeddings for environments and tasks. In this stage, we use only one demonstration from each $(\varepsilon, \tau)$ pair to fine-tune the embedding and find that our network is able to generalize to new environment or/and task.

Concretely, we randomly initialize the 10 new environment embeddings and the 10 new task embeddings for additional learning. In the transfer setting 2, we sample only one expert trajectory as demonstration data for each $(\varepsilon, \tau)$ pair in the upper right and lower left quadrant. In the transfer settings 3, we sample only one expert trajectory as demonstration data for each $(\varepsilon, \tau)$ pair in the lower right quadrant. Following the same routine Algorithm 1, we train the embeddings for 10000 iterations and then test the performance of models on the entire matrix of $(\varepsilon, \tau)$ pairs. The result is shown as Figure 8. Besides what we have mentioned in the main text, we plot a more visually discernible success rate matrices as Figure 8 (a) and (b). We observe that in both cases, transfer learning across the task axis is easier comparing to the environment axis, given the results.

## 5.5    An extreme studies about the effectiveness of environment embeddings.

As mentioned in the main text, to study the effectiveness of the environment embedding, we run an additional experiment as a sanity check. In this setting, we made agent's observation window size

**(a) 10 Train and 90 Test**

|  | env_0 | env_1 | env_2 | env_3 | env_4 | env_5 | env_6 | env_7 | env_8 | env_9 |
|---|---|---|---|---|---|---|---|---|---|---|
| ('R', 'B') | 0.90 | 0.10 | 0.20 | 0.10 | 0.10 | 0.30 | 0.40 | 0.20 | 0.10 | 0.00 |
| ('B', 'G') | 0.10 | 1.00 | 0.00 | 0.00 | 0.00 | 0.20 | 0.30 | 0.10 | 0.00 | 0.10 |
| ('G', 'O') | 0.10 | 0.10 | 1.00 | 0.10 | 0.00 | 0.20 | 0.00 | 0.00 | 0.30 | 0.00 |
| ('O', 'P') | 0.00 | 0.00 | 0.00 | 1.00 | 0.00 | 0.00 | 0.20 | 0.00 | 0.20 | 0.20 |
| ('P', 'R') | 0.40 | 0.00 | 0.00 | 0.20 | 1.00 | 0.30 | 0.00 | 0.00 | 0.30 | 0.10 |
| ('R', 'G') | 0.30 | 0.00 | 0.00 | 0.00 | 0.10 | 0.90 | 0.20 | 0.30 | 0.10 | 0.00 |
| ('B', 'O') | 0.10 | 0.30 | 0.10 | 0.00 | 0.00 | 0.10 | 0.90 | 0.10 | 0.00 | 0.10 |
| ('G', 'P') | 0.20 | 0.10 | 0.10 | 0.00 | 0.10 | 0.10 | 0.10 | 1.00 | 0.00 | 0.10 |
| ('O', 'R') | 0.10 | 0.00 | 0.00 | 0.00 | 0.10 | 0.00 | 0.20 | 0.00 | 1.00 | 0.00 |
| ('P', 'B') | 0.40 | 0.20 | 0.10 | 0.00 | 0.20 | 0.10 | 0.10 | 0.00 | 0.00 | 0.90 |

**(b) 20 Train and 80 Test**

|  | env_0 | env_1 | env_2 | env_3 | env_4 | env_5 | env_6 | env_7 | env_8 | env_9 |
|---|---|---|---|---|---|---|---|---|---|---|
| ('R', 'B') | 1.00 | 0.10 | 1.00 | 0.20 | 0.10 | 0.00 | 0.60 | 0.10 | 0.20 | 0.20 |
| ('B', 'G') | 0.50 | 0.30 | 0.60 | 0.20 | 0.10 | 0.50 | 0.90 | 0.30 | 0.50 | 0.90 |
| ('G', 'O') | 0.10 | 0.10 | 0.50 | 0.30 | 0.20 | 0.10 | 0.30 | 0.40 | 1.00 | 1.00 |
| ('O', 'P') | 0.70 | 1.00 | 0.50 | 0.30 | 1.00 | 0.90 | 1.00 | 0.80 | 0.50 | 0.90 |
| ('P', 'R') | 0.30 | 0.40 | 1.00 | 0.90 | 0.20 | 0.60 | 0.30 | 0.90 | 0.70 | 0.60 |
| ('R', 'G') | 0.30 | 0.30 | 0.20 | 0.20 | 0.10 | 0.40 | 0.90 | 0.10 | 1.00 | 0.80 |
| ('B', 'O') | 0.50 | 0.30 | 0.60 | 0.30 | 0.50 | 0.30 | 0.90 | 0.20 | 0.30 | 0.90 |
| ('G', 'P') | 0.40 | 0.20 | 0.10 | 0.10 | 0.30 | 1.00 | 0.30 | 0.10 | 0.10 | 0.20 |
| ('O', 'R') | 0.40 | 0.00 | 0.80 | 0.00 | 0.00 | 0.00 | 0.30 | 0.10 | 0.60 | 0.50 |
| ('P', 'B') | 0.10 | 0.30 | 0.30 | 0.90 | 0.30 | 0.60 | 0.50 | 0.40 | 0.30 | 1.00 |

**(c) 30 Train and 70 Test**

|  | env_0 | env_1 | env_2 | env_3 | env_4 | env_5 | env_6 | env_7 | env_8 | env_9 |
|---|---|---|---|---|---|---|---|---|---|---|
| ('R', 'B') | 0.60 | 1.00 | 0.80 | 0.40 | 1.00 | 0.30 | 1.00 | 0.60 | 0.60 | 0.50 |
| ('B', 'G') | 0.50 | 0.10 | 0.90 | 0.30 | 0.40 | 0.90 | 0.80 | 0.90 | 0.80 | 0.90 |
| ('G', 'O') | 1.00 | 0.80 | 0.90 | 0.90 | 0.10 | 0.50 | 0.70 | 0.70 | 0.80 | 0.80 |
| ('O', 'P') | 1.00 | 0.50 | 1.00 | 0.90 | 0.30 | 0.80 | 0.90 | 0.30 | 0.40 | 0.90 |
| ('P', 'R') | 1.00 | 1.00 | 0.70 | 1.00 | 0.70 | 1.00 | 0.90 | 0.50 | 0.90 | 0.80 |
| ('R', 'G') | 0.90 | 0.50 | 0.90 | 1.00 | 0.60 | 0.40 | 0.90 | 0.50 | 0.60 | 0.80 |
| ('B', 'O') | 0.60 | 0.60 | 0.70 | 0.40 | 0.40 | 1.00 | 0.90 | 0.90 | 0.40 | 1.00 |
| ('G', 'P') | 1.00 | 1.00 | 0.90 | 0.90 | 0.50 | 0.60 | 0.70 | 0.60 | 1.00 | 1.00 |
| ('O', 'R') | 0.10 | 0.90 | 0.60 | 0.70 | 1.00 | 0.70 | 0.90 | 0.20 | 1.00 | 0.40 |
| ('P', 'B') | 0.60 | 0.40 | 0.90 | 0.60 | 0.40 | 0.50 | 1.00 | 0.80 | 0.30 | 1.00 |

**(d) 40 Train and 60 Test**

|  | env_0 | env_1 | env_2 | env_3 | env_4 | env_5 | env_6 | env_7 | env_8 | env_9 |
|---|---|---|---|---|---|---|---|---|---|---|
| ('R', 'B') | 0.90 | 0.90 | 0.80 | 1.00 | 0.90 | 1.00 | 1.00 | 0.90 | 1.00 | 0.70 |
| ('B', 'G') | 0.90 | 1.00 | 1.00 | 0.70 | 0.90 | 1.00 | 1.00 | 0.80 | 0.90 | 0.50 |
| ('G', 'O') | 0.90 | 1.00 | 0.80 | 0.80 | 0.40 | 1.00 | 1.00 | 1.00 | 1.00 | 1.00 |
| ('O', 'P') | 0.90 | 1.00 | 0.90 | 1.00 | 0.90 | 1.00 | 0.90 | 0.70 | 0.70 | 1.00 |
| ('P', 'R') | 1.00 | 0.90 | 0.60 | 0.70 | 0.70 | 0.90 | 0.80 | 1.00 | 0.90 | 0.90 |
| ('R', 'G') | 0.90 | 1.00 | 0.90 | 0.70 | 0.90 | 0.80 | 1.00 | 0.80 | 1.00 | 0.70 |
| ('B', 'O') | 0.80 | 1.00 | 1.00 | 1.00 | 0.90 | 1.00 | 1.00 | 1.00 | 1.00 | 0.80 |
| ('G', 'P') | 0.90 | 0.80 | 1.00 | 0.80 | 0.70 | 1.00 | 1.00 | 0.90 | 0.80 | 0.70 |
| ('O', 'R') | 1.00 | 0.90 | 0.90 | 0.60 | 0.90 | 1.00 | 0.90 | 0.90 | 0.80 | 1.00 |
| ('P', 'B') | 0.80 | 1.00 | 0.80 | 1.00 | 1.00 | 1.00 | 0.80 | 1.00 | 1.00 | 0.80 |

Figure 4: Average test success rate on each environment-task combination. Blue grids represent seen combinations and red grids represent unseen combinations

to be 1, which made agent only capable of seeing itself and the location of treasures on the map, without any knowledge about the maze. We denote this agent as a "blind" agent. Therefore, such a agent would need to remember the structure of the maze to perform well under this circumstance. We follow our original imitation training process as well as evaluation process and tested three representative methods in this setting, and plot the results as Table 5. As we have expected, we observe that algorithms such as MTL which do not distinguish between environments would fail severely. It could still success in some cases such as the treasures are generated at the same room as the agent, or very close by. With the additional environment embedding, a simple algorithm such as MLP could significantly outperforms this degenerated multi-task model. In addition, SYNPO can achieve almost as good as it was in the normal circumstance, demonstrating its strong capability in memorizing the environment.

## Footnotes

†On leave from University of Southern California (feisha@usc.edu).

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

**(a) 50 Train and 50 Test**

**(b) 60 Train and 40 Test**

**(c) 70 Train and 30 Test**

**(d) 80 Train and 20 Test**

Figure 5: Average test success rate on each environment-task combination. Blue grids represent seen combinations and red grids represent unseen combinations

Table 5: Performance of SynPo, MTL and MLP on GRIDWORLD (SEEN/UNSEEN=144/256) with window size = 0. All algorithms trained are trained using three random seeds and reported with mean and std.

| Method | MLP | MTL | SYNPO |
|---|---|---|---|
| AvgSR. (SEEN) | $56.8 \pm 0.9\%$ | $16.4 \pm 0.4\%$ | $\mathbf{80.9 \pm 1.5}$ % |
| AvgSR. (UNSEEN) | $51.8 \pm 1.7\%$ | $6.1 \pm 0.2\%$ | $\mathbf{76.8 \pm 1.4\%}$ |

for multi-task and multi-robot transfer. In *Robotics and Automation (ICRA), 2017 IEEE International Conference on*, pages 2169–2176. IEEE, 2017.

[3] K. He, X. Zhang, S. Ren, and J. Sun. Deep residual learning for image recognition. In *Proceedings of the IEEE conference on computer vision and pattern recognition*, pages 770–778, 2016.

[4] D. P. Kingma and J. Ba. Adam: A method for stochastic optimization. *arXiv preprint arXiv:1412.6980*, 2014.

[5] E. Kolve, R. Mottaghi, D. Gordon, Y. Zhu, A. Gupta, and A. Farhadi. Ai2-thor: An interactive 3d environment for visual ai. *CoRR*, abs/1712.05474, 2017.

[6] V. Mnih, K. Kavukcuoglu, D. Silver, A. A. Rusu, J. Veness, M. G. Bellemare, A. Graves, M. A. Riedmiller, A. Fidjeland, G. Ostrovski, S. Petersen, C. Beattie, A. Sadik, I. Antonoglou, H. King, D. Kumaran,

Figure 6: Case study for a situation when the ratio of # of combinations seen and the total is 0.2

D. Wierstra, S. Legg, and D. Hassabis. Human-level control through deep reinforcement learning. *Nature*, 518:529–533, 2015.

[7] J. Schulman, P. Moritz, S. Levine, M. Jordan, and P. Abbeel. High-dimensional continuous control using generalized advantage estimation. *arXiv preprint arXiv:1506.02438*, 2015.

[8] J. Schulman, F. Wolski, P. Dhariwal, A. Radford, and O. Klimov. Proximal policy optimization algorithms. *arXiv preprint arXiv:1707.06347*, 2017.

[9] Y. Zhu, D. Gordon, E. Kolve, D. Fox, L. Fei-Fei, A. Gupta, R. Mottaghi, and A. Farhadi. Visual semantic planning using deep successor representations. In *Proceedings of the IEEE International Conference on Computer Vision*, volume 2, page 7, 2017.

[10] Y. Zhu, R. Mottaghi, E. Kolve, J. J. Lim, A. Gupta, L. Fei-Fei, and A. Farhadi. Target-driven visual navigation in indoor scenes using deep reinforcement learning. *2017 IEEE International Conference on Robotics and Automation (ICRA)*, pages 3357–3364, 2017.

(a) AvgSR. on SEEN Split

(b) AvgSR. on UNSEEN Split

(c) AvgReward on SEEN Split

(d) AvgReward on UNSEEN Split

Figure 7: **Results of "A blind agent scenario" on GRIDWORLD with window size of 0.** (a)-(b): Comparison between average success rate (ASR.) of algorithms on seen split and unseen split. (c)-(d): Comparison between average accumulated reward (AvgReward.) of algorithms in each episode on seen split and unseen split. Results are reported on the setting with $|\mathcal{E}| = 20$ *and* $|\mathcal{T}| = 20$. For each intermediate performance, we sample 100 $(\varepsilon, \tau)$ combinations and test one configuration to evaluate the performances. We evaluate models trained with 3 random seeds and report results in terms of the mean AvgSR and its standard deviation.

**(a) Transfer Setting 2**

| | Env_0 | Env_1 | Env_2 | Env_3 | Env_4 | Env_5 | Env_6 | Env_7 | Env_8 | Env_9 | Env_10 | Env_11 | Env_12 | Env_13 | Env_14 | Env_15 | Env_16 | Env_17 | Env_18 | Env_19 |
|---|---|---|---|---|---|---|---|---|---|---|---|---|---|---|---|---|---|---|---|---|
| ('R', 'B') | 0.90 | 1.00 | 1.00 | 0.90 | 0.90 | 1.00 | 1.00 | 0.80 | 1.00 | 0.90 | 0.30 | 0.20 | 0.10 | 0.50 | 0.00 | 0.40 | 0.20 | 0.10 | 0.50 | 0.00 |
| ('B', 'G') | 1.00 | 0.90 | 1.00 | 1.00 | 1.00 | 0.90 | 1.00 | 0.80 | 1.00 | 1.00 | 0.00 | 0.30 | 0.50 | 0.20 | 0.10 | 0.30 | 0.40 | 0.20 | 0.00 | 0.00 |
| ('G', 'O') | 0.90 | 1.00 | 1.00 | 0.80 | 1.00 | 1.00 | 1.00 | 0.90 | 1.00 | 0.90 | 0.20 | 0.40 | 0.30 | 0.30 | 0.20 | 0.50 | 0.20 | 0.20 | 0.10 | 0.00 |
| ('O', 'P') | 0.90 | 0.90 | 1.00 | 0.90 | 0.90 | 0.80 | 0.80 | 0.90 | 0.80 | 1.00 | 0.40 | 0.30 | 0.10 | 0.20 | 0.20 | 0.40 | 0.10 | 0.00 | 0.00 | 0.10 |
| ('P', 'R') | 1.00 | 0.90 | 1.00 | 1.00 | 1.00 | 0.90 | 1.00 | 1.00 | 1.00 | 0.70 | 0.10 | 0.10 | 0.10 | 0.10 | 0.20 | 0.30 | 0.40 | 0.00 | 0.10 | 0.00 |
| ('R', 'G') | 0.90 | 1.00 | 1.00 | 1.00 | 0.80 | 1.00 | 1.00 | 1.00 | 1.00 | 1.00 | 0.40 | 0.40 | 0.10 | 0.20 | 0.10 | 0.50 | 0.30 | 0.00 | 0.00 | 0.00 |
| ('B', 'O') | 1.00 | 0.90 | 1.00 | 0.90 | 0.90 | 1.00 | 0.90 | 1.00 | 0.90 | 0.80 | 0.20 | 0.20 | 0.30 | 0.20 | 0.40 | 0.40 | 0.50 | 0.10 | 0.00 | 0.10 |
| ('G', 'P') | 0.90 | 1.00 | 0.90 | 0.90 | 0.80 | 1.00 | 0.80 | 0.90 | 0.80 | 0.80 | 0.10 | 0.30 | 0.40 | 0.50 | 0.10 | 0.50 | 0.00 | 0.20 | 0.00 | 0.00 |
| ('O', 'R') | 1.00 | 0.90 | 1.00 | 0.80 | 0.90 | 0.90 | 1.00 | 1.00 | 1.00 | 1.00 | 0.50 | 0.10 | 0.10 | 0.20 | 0.20 | 0.50 | 0.40 | 0.20 | 0.10 | 0.20 |
| ('P', 'B') | 1.00 | 0.90 | 0.90 | 0.90 | 1.00 | 1.00 | 1.00 | 0.90 | 1.00 | 1.00 | 0.30 | 0.20 | 0.40 | 0.20 | 0.50 | 0.30 | 0.20 | 0.20 | 0.30 | 0.00 |
| ('R', 'O') | 0.50 | 0.30 | 0.50 | 0.50 | 0.40 | 0.40 | 0.70 | 0.30 | 0.60 | 0.60 | 0.20 | 0.10 | 0.20 | 0.20 | 0.10 | 0.20 | 0.20 | 0.00 | 0.00 | 0.10 |
| ('B', 'P') | 0.30 | 0.50 | 0.40 | 0.10 | 0.50 | 0.20 | 0.30 | 0.50 | 0.60 | 0.70 | 0.10 | 0.00 | 0.20 | 0.10 | 0.00 | 0.10 | 0.20 | 0.00 | 0.10 | 0.10 |
| ('G', 'R') | 0.40 | 0.80 | 0.80 | 0.80 | 0.60 | 0.50 | 0.80 | 0.70 | 0.60 | 0.80 | 0.20 | 0.20 | 0.10 | 0.10 | 0.20 | 0.20 | 0.50 | 0.20 | 0.00 | 0.00 |
| ('O', 'B') | 0.60 | 0.50 | 0.50 | 0.60 | 0.50 | 0.30 | 0.90 | 0.60 | 0.50 | 0.60 | 0.20 | 0.10 | 0.20 | 0.10 | 0.10 | 0.20 | 0.40 | 0.00 | 0.10 | 0.10 |
| ('P', 'G') | 0.50 | 0.50 | 0.40 | 0.50 | 0.60 | 0.60 | 0.40 | 0.50 | 0.60 | 0.80 | 0.10 | 0.20 | 0.40 | 0.70 | 0.10 | 0.10 | 0.00 | 0.10 | 0.10 | 0.10 |
| ('R', 'P') | 0.30 | 0.30 | 0.50 | 0.70 | 0.50 | 0.80 | 0.30 | 0.50 | 0.50 | 0.60 | 0.10 | 0.20 | 0.20 | 0.10 | 0.10 | 0.10 | 0.30 | 0.00 | 0.10 | 0.00 |
| ('B', 'R') | 0.50 | 0.50 | 0.20 | 0.20 | 0.30 | 0.50 | 0.40 | 0.60 | 0.50 | 0.70 | 0.00 | 0.10 | 0.20 | 0.20 | 0.00 | 0.20 | 0.30 | 0.00 | 0.00 | 0.20 |
| ('G', 'B') | 0.30 | 0.30 | 0.30 | 0.40 | 0.10 | 0.50 | 0.20 | 0.20 | 0.20 | 0.40 | 0.00 | 0.00 | 0.10 | 0.00 | 0.10 | 0.00 | 0.10 | 0.00 | 0.10 | 0.10 |
| ('O', 'G') | 0.60 | 0.70 | 0.50 | 0.40 | 0.50 | 0.70 | 0.40 | 0.80 | 0.60 | 0.40 | 0.10 | 0.20 | 0.20 | 0.20 | 0.30 | 0.30 | 0.40 | 0.20 | 0.10 | 0.00 |
| ('P', 'O') | 0.70 | 0.60 | 0.70 | 0.70 | 0.40 | 0.80 | 0.40 | 0.40 | 0.60 | 0.60 | 0.00 | 0.10 | 0.30 | 0.10 | 0.10 | 0.30 | 0.30 | 0.00 | 0.10 | 0.00 |

(a) Transfer Setting 2

**(b) Transfer Setting 3**

| | Env_0 | Env_1 | Env_2 | Env_3 | Env_4 | Env_5 | Env_6 | Env_7 | Env_8 | Env_9 | Env_10 | Env_11 | Env_12 | Env_13 | Env_14 | Env_15 | Env_16 | Env_17 | Env_18 | Env_19 |
|---|---|---|---|---|---|---|---|---|---|---|---|---|---|---|---|---|---|---|---|---|
| ('R', 'B') | 1.00 | 1.00 | 1.00 | 1.00 | 0.90 | 1.00 | 1.00 | 0.90 | 0.80 | 0.80 | 0.40 | 0.20 | 0.30 | 0.20 | 0.10 | 0.40 | 0.10 | 0.40 | 0.10 | 0.00 |
| ('B', 'G') | 1.00 | 1.00 | 0.80 | 0.80 | 0.90 | 1.00 | 1.00 | 0.80 | 0.80 | 1.00 | 0.20 | 0.30 | 0.20 | 0.30 | 0.10 | 0.20 | 0.50 | 0.00 | 0.00 | 0.10 |
| ('G', 'O') | 1.00 | 0.90 | 0.90 | 0.90 | 0.80 | 0.90 | 1.00 | 0.90 | 0.80 | 1.00 | 0.30 | 0.20 | 0.10 | 0.20 | 0.20 | 0.30 | 0.20 | 0.20 | 0.10 | 0.00 |
| ('O', 'P') | 1.00 | 0.90 | 1.00 | 1.00 | 1.00 | 0.70 | 0.70 | 0.70 | 1.00 | 0.90 | 0.20 | 0.30 | 0.40 | 0.30 | 0.20 | 0.40 | 0.20 | 0.00 | 0.00 | 0.20 |
| ('P', 'R') | 0.90 | 0.90 | 0.70 | 0.80 | 1.00 | 0.90 | 0.90 | 0.80 | 1.00 | 0.90 | 0.20 | 0.10 | 0.10 | 0.20 | 0.00 | 0.50 | 0.10 | 0.00 | 0.10 | 0.00 |
| ('R', 'G') | 0.90 | 1.00 | 0.90 | 0.90 | 0.80 | 0.90 | 0.80 | 0.80 | 1.00 | 0.80 | 0.50 | 0.20 | 0.20 | 0.00 | 0.10 | 0.10 | 0.20 | 0.30 | 0.30 | 0.20 |
| ('B', 'O') | 0.90 | 0.90 | 0.90 | 0.80 | 0.90 | 0.80 | 0.90 | 0.80 | 0.80 | 0.90 | 0.50 | 0.10 | 0.30 | 0.20 | 0.10 | 0.30 | 0.20 | 0.10 | 0.10 | 0.10 |
| ('G', 'P') | 0.90 | 1.00 | 1.00 | 0.90 | 1.00 | 0.80 | 0.90 | 0.90 | 0.80 | 1.00 | 0.30 | 0.10 | 0.10 | 0.10 | 0.00 | 0.10 | 0.30 | 0.10 | 0.00 | 0.10 |
| ('O', 'R') | 0.90 | 0.90 | 1.00 | 1.00 | 0.90 | 0.90 | 0.90 | 1.00 | 1.00 | 0.90 | 0.20 | 0.40 | 0.30 | 0.10 | 0.20 | 0.30 | 0.30 | 0.20 | 0.10 | 0.30 |
| ('P', 'B') | 1.00 | 1.00 | 1.00 | 0.90 | 1.00 | 1.00 | 0.90 | 0.90 | 1.00 | 1.00 | 0.10 | 0.20 | 0.30 | 0.30 | 0.20 | 0.50 | 0.20 | 0.00 | 0.30 | 0.20 |
| ('R', 'O') | 0.40 | 0.40 | 0.30 | 0.50 | 0.40 | 0.50 | 0.40 | 0.30 | 0.00 | 0.30 | 0.20 | 0.30 | 0.10 | 0.10 | 0.00 | 0.40 | 0.30 | 0.20 | 0.20 | 0.10 |
| ('B', 'P') | 0.50 | 0.30 | 0.30 | 0.30 | 0.80 | 0.30 | 0.50 | 0.40 | 0.40 | 0.30 | 0.10 | 0.00 | 0.20 | 0.20 | 0.20 | 0.40 | 0.30 | 0.00 | 0.00 | 0.00 |
| ('G', 'R') | 0.80 | 0.20 | 0.70 | 0.50 | 0.60 | 0.60 | 0.50 | 0.70 | 0.70 | 0.70 | 0.00 | 0.10 | 0.20 | 0.00 | 0.00 | 0.20 | 0.40 | 0.20 | 0.00 | 0.00 |
| ('O', 'B') | 0.20 | 0.40 | 0.50 | 0.40 | 0.20 | 0.60 | 0.30 | 0.40 | 0.40 | 0.40 | 0.20 | 0.10 | 0.00 | 0.20 | 0.10 | 0.10 | 0.00 | 0.00 | 0.10 | 0.00 |
| ('P', 'G') | 0.50 | 0.60 | 0.50 | 0.40 | 0.20 | 0.90 | 0.80 | 0.20 | 0.80 | 0.70 | 0.20 | 0.00 | 0.10 | 0.10 | 0.10 | 0.10 | 0.30 | 0.10 | 0.10 | 0.40 |
| ('R', 'P') | 0.40 | 0.20 | 0.40 | 0.40 | 0.20 | 0.20 | 0.40 | 0.40 | 0.40 | 0.40 | 0.20 | 0.10 | 0.20 | 0.10 | 0.10 | 0.20 | 0.20 | 0.20 | 0.10 | 0.20 |
| ('B', 'R') | 0.50 | 0.30 | 0.50 | 0.40 | 0.20 | 0.40 | 0.30 | 0.30 | 0.50 | 0.30 | 0.10 | 0.00 | 0.10 | 0.00 | 0.30 | 0.10 | 0.10 | 0.00 | 0.00 | 0.10 |
| ('G', 'B') | 0.30 | 0.10 | 0.30 | 0.30 | 0.40 | 0.50 | 0.10 | 0.30 | 0.30 | 0.20 | 0.10 | 0.10 | 0.00 | 0.00 | 0.00 | 0.10 | 0.00 | 0.00 | 0.10 | 0.00 |
| ('O', 'G') | 0.80 | 0.90 | 0.70 | 0.80 | 0.60 | 0.60 | 0.40 | 0.70 | 0.70 | 0.90 | 0.00 | 0.40 | 0.10 | 0.40 | 0.00 | 0.70 | 0.30 | 0.00 | 0.00 | 0.10 |
| ('P', 'O') | 0.00 | 0.10 | 0.30 | 0.30 | 0.50 | 0.40 | 0.40 | 0.10 | 0.40 | 0.20 | 0.10 | 0.00 | 0.10 | 0.20 | 0.10 | 0.40 | 0.00 | 0.00 | 0.10 | 0.40 |

(b) Transfer Setting 3

Figure 8: Visualizing the effectiveness transferring. Average success rates are marked in the grid (more visually discernible plots are in the Suppl. Materials). The purple cells are from $Q$ set and red cells represents the rest. The darker the color is, the better the corresponding performance.

Figure 9: Visualization of the environments we used for GRIDWORLD experiments. The environments we used are very different against each other, thus placed a substantial challenge for agent to generalize. (Note that agent's and objects' positions are randomized.)