[Reviews · NeurIPS 2018]

Reviewer 1



Abstract: Would remove the capitalized (ENV, TASK) from the text. Introduction: Line 39-46, the actual terminology used - policy basis, embeddings, etc are not at all described so the description here is quite confusing. Related work: Other things to mention Multi-task multi-robot transfer -(Devin et al, (although mentioned later, good to mention in related work)), Invariant feature spaces transfer (Gupta et al), work by bou ammar on transfer learning, progressive nets (rusu et al). Section 3.1 - very nice problem definition in general. A careful consideration of the three scenarios mentioned in the introduction would be good to have here. In Fig 2, what is a task descriptor and what is an environment descriptor . Section 3.2 - Key idea seems to be a parameterization of the policy via a bilinear mapping with state features, action features and a metric U which depends on e_env, e_task. Reward is also predicted but with different coefficients. This is similar to successor features. Section 3.3 In order to impose more structure since parameters of alpha and inputs are all being optimized, a disentanglement loss is being optimized. The idea is very simple - discriminability of env or task from the state-action representation. It has been used effectively in a number of other scenarios - eg unsupervised exploration and is a good choice here. Section 3.5 A slightly more in-depth consideration of the fine-tuning setup would be helpful here. Experiments are interesting and helpful, the ablation study is well done and a number of meaningful baselines are compared with. Figure 6 is especially nice in understanding transfer capabilities. Table 3 empirical performance seems a bit weak overall. Any reasoning behind this? Overall I think this work is clearly motivated, builds nicely on prior work and does a good job of experiments. I would recommend an accept.

Reviewer 2



The paper focuses on the key problem of transfer learning policies that can generalize across a range of task and environments despite being trained on a sparse subset. The paper forces disentanglement between tasks and environments by introducing discriminative losses. However, the paper uses behavior cloning to learn these policies rather than any of the DeepRL approaches. The paper tests their performance against other baseline approaches on a set of GridWorld environments and tasks as well as more challenging 3D THOR environment and tasks and show significant improvement in performance. The paper also does an ablation study to find which components of their system is more important than the others with Disentanglement losses being significantly important. The paper clearly distinguishes tasks and environemnts and identifies three progressively difficult settings for adaptation and transfer. It proposes learning separate task and environment embeddings that can be composed together to give a policy (some related ideas in [1] as is one of the baselines). It's a little disappointment to see behavior cloning being used instead of RL. It's easy to say that this can be integrated with RL algorithms but it's not as easy as it looks. Gradients are much worse in RL and the reward signal a lot noisier. So learning would be a lot harder and might not work at all. It would have been useful to see some example at least. And it seems like you do need one-shot demonstration in a new task environment and it's not clear until end of section 3.5 and should have been mentioned earlier and more clearly. Figure 6 is hard to see (but understandable given NIPS page limit). Although the cross pair study and the marked difference Is quite interesting. Not sure if the baselines have been carefully tuned. One would expect ModuleNet to perform atleast a bit better than a plain MLP but there seems no difference. It's also unclear how many runs are the results averaged over, although this being supervised learning, probably not that big of a difference. Hopefully the source code for the experiments will be released. [1]: https://arxiv.org/abs/1609.07088

Reviewer 3



The paper presents a framework for learning to transfer across environments as well as task. The framework consists of separate environment and task descriptor which generate embeddings and a policy network and reward function that takes environment, task, state and action embedding as input and outputs policy and reward through a look up table. In order to transfer to unseen environment and task the look up table for the environment and task is updated which enables fine tuning with just a few demonstration. The look up table acts as an attention mechanism which is updated for new environment and task. It is difficult to interpret the results of visualization from t-SNE of environment embedding. Additionally, it would be interesting to see saliency or activation maximization based method applied to the task descriptor to see whether the saliency of the network is consistent. Instead of average reward, it would be interesting to see the reward mean and variance for different algorithms. It would be interesting to see how the method compares to the meta-learning approach. Minor Corrections: Line 148: typo "relies" Line 171: typo "state" Update: The authors did a good job in their response and also cleared some misunderstandings I had with the paper. I've changed my score to an accept. One of the problems I feel that is still not addressed properly in the paper is the interpretability (t-SNE analysis) but overall experiments and ablation studies are thorough.